# Salvage External Beam Radiotherapy after Incomplete Transarterial Chemoembolization for Hepatocellular Carcinoma: A Meta-Analysis and Systematic Review

**DOI:** 10.3390/medicina57101000

**Published:** 2021-09-22

**Authors:** Dae Sik Yang, Sunmin Park, Chai Hong Rim, Won Sup Yoon, In-Soo Shin, Han Ah Lee

**Affiliations:** 1Department of Radiation Oncology, Korea University Medical College, Seoul 02841, Korea; irionmphage@korea.ac.kr (D.S.Y.); irionyws@korea.ac.kr (W.S.Y.); 2Department of Radiation Oncology, Korea University Guro Hospital, Seoul 08308, Korea; 3Department of Radiation Oncology, Korea University Ansan Hospital, Ansan 15355, Korea; sunmini815@gmail.com; 4Graduate School of Education, AI Convergence Education, Dongguk University, Seoul 04620, Korea; s9065031@dongguk.edu; 5Department of Gastroenterology, Anam Hospital, Korea University Medical College, Seoul 02841, Korea; amelia86@naver.com

**Keywords:** radiotherapy, external beam radiotherapy, salvage, hepatocellular carcinoma, transarterial chemoembolization

## Abstract

*Background and objective*: Although transarterial chemoembolization (TACE) has been the commonest local modality for hepatocellular carcinoma (HCC), incomplete repsonse occurs especially for tumors with a large size or difficult tumor accessment. The present meta-analysis assessed the efficacy and feasibility of external beam radiotherapy (EBRT) as a salvage modality after incomplete TACE. *Materials and Methods*: We systematically searched the PubMed, Embase, Medline, and Cochrane databases. The primary endpoint was overall survival (OS), and the secondary endpoints included the response ratem toxicity of grade 3, and local control. *Results*: Twelve studies involving 757 patients were included; the median of portal vein thrombosis rate was 25%, and the pooled median of tumor size was 5.8 cm. The median prescribed dose ranged from 37.3 to 150 Gy (pooled median: 54 Gy in *EQD2). The pooled one- and two-year OS rates were 72.3% (95% confidence interval (CI): 60.2–81.9%) and 50.5% (95% CI: 35.6–65.4%), respectively; the pooled response and local control rates were 72.2% (95% CI: 65.4–78.1%) and 86.6 (95% CI: 80.1–91.2%) respectively. The pooled rates of grade ≥3 gastrointestinal toxicity, radiation-induced liver disease, hepatotoxicity, and hematotoxicity were 4.1%, 3.5%, 5.7%, and 4.9%, respectively. Local control was not correlated with intrahepatic (*p* = 0.6341) or extrahepatic recurrences (*p* = 0.8529) on meta-regression analyses. *Conclusion*: EBRT was feasible and efficient in regard to tumor response and control; after incomplete TACE. Out-field recurrence, despite favorable local control, necessitates the combination of EBRT with systemic treatments. *Equivalent dose in 2 Gy per fraction scheme.

## 1. Introduction

Until a few decades ago, treatment of liver cancer was limited to surgical approaches. However, liver cirrhosis or decompensation and the local invasion of major anatomical structures often hinder safe surgery. Approximately 70% of all patients with hepatocellular carcinoma (HCC) primarily undergo non-surgical treatment. Transarterial chemoembolization (TACE) is the most commonly applied non-surgical local modality [1,2].

The oncologic benefit of TACE for unresectable HCC is well-established in previous randomized trials [3,4]. For early-stage HCC (i.e., ≤3 cm without vascular invasion), the efficacy of TACE is comparable to that of other local modalities such as surgery or ablation [5,6]. The main limitation of TACE is that complete remission and sustained local control are rarely achieved for large or locally advanced tumors. In a recent meta-analysis, the two-year progression-free survival after TACE for HCC was 24%, with an overwhelming majority of patients experiencing recurrence within one year [7].

Currently, no standard modality has been established for patients with residual tumors or incomplete TACE. A common approach in such cases is to repeat TACE. However, such an approach has concern of post-embolization syndromes [7]. Moreover, it is difficult to overcome anatomical hindrances for locally advanced tumors, such as the existence of collateral vasculature or the presence of less responsive avascularized tumors areas [8]. Sorafenib administration is a conventional option; however, considering the low tumor response rate to this agent [9,10] and the persistence of residual disease after TACE for locally advanced tumors, therapeutic outcomes might not be satisfactory.

External beam radiotherapy (EBRT) has recently emerged as a treatment for HCC and has several advantages. Modern EBRT with computed tomography planning can encompass large tumors with a homogeneous dose distribution [11,12]. The response rates of EBRT range from 50% to 80%, even for tumors with major vessel invasion [13,14]. For relatively small tumors, EBRT could be applied as a comparable surrogate to radiofrequency ablation, especially for tumors with a large size or difficult locations [15,16]. According to a recent meta-analysis, the one- and two-year local control rates for tumors measuring up to 5 cm that were treated with stereotactic body radiotherapy (SBRT) were as high as 91% and 87%, respectively [17].

Given these efficacies, several researchers have used EBRT as a salvage modality for patients with incomplete TACE. In this study, we will integrate data from previous studies to investigate the efficacy and feasibility of EBRT as a salvage option for TACE.

## 2. Materials and Methods

### 2.1. Study Selection

The present meta-analysis and systematic review were conducted to answer the following PICO question: “Is EBRT a feasible and efficacious option as a salvage modality after incomplete TACE?” We adhered to the guidance of the Preferred Reporting Items for Systematic Reviews and Meta-Analyses (PRISMA) and referenced the Cochrane Handbook version 6.2 for the methodological regard. We systematically searched four databases—PubMed, Embase, Medline, and Cochrane databases, as recommended by the Cochrane Handbook [18]—for entries up to 25 January 2021. We used the following search term: “(incomplete OR residual OR recurrent) AND (TACE or transarterial) AND (radiotherapy OR radiation) AND (liver OR HCC OR hepatocellular)“. The related articles in the reference lists were also searched; no language or period restriction was applied. Conference abstracts were not included owing to their lack of information regarding clinical prognosticators.

The following criteria were used: (1) clinical studies involving patients who underwent EBRT owing to incomplete TACE or residual lesions thereafter; (2) at least 10 relevant patients; and (3) provision of the primary endpoint. For our study, the primary endpoint was overall survival (OS), while the secondary endpoints included the response rate as well as the pattern and rate of failure (e.g., in-field, intrahepatic, or extrahepatic failure), and grade ≥3 complications. Grade 5 complications were subjectively assessed.

After our initial search, articles of irrelevant types (including reviews, editorials, letters, and conference abstracts) and duplicates were excluded based on their titles and citations. Two independent reviewers consequently screened the abstracts to filter out studies with irrelevant subjects or formats. Full-text reviews were finally performed by two independent reviewers to identify studies that fully met the inclusion criteria. Multiple studies from the same institution were included, only if there were no (or negligible) overlapping data. If substantial data overlapping existed, the following criteria were used for selection and prioritized in numerical order: (1) studies with more relevant subjects (e.g., studies solely regarding patients underwent EBRT after incomplete TACE were preferred, rather than those having such patients as a subgroup); (2) a larger number of patients; (3) the publication dates are more recent, if the numbers of recruited patients in studies are grossly similar.

### 2.2. Data Extraction

Data extraction was performed by two independent researchers using a pre-standardized form that collected the following: (1) general information including author names, affiliations, countries, and publication years; (2) clinical data including the number of patients, age, etiology, sex, performance status, alpha-fetoprotein (AFP), portal vein thrombosis (PVT), Child-Pugh class, tumor size, and EBRT dose; and (3) clinical outcomes including the OS rates, response rate, pattern of failure, and grade ≥3 toxicity rate. Any missing numerical OS rates were estimated from descriptive graphs, as available. Toxicities were classified into four categories—gastrointestinal (such as gastritis or duodenitis, bowel bleeding, and abdominal pain), classic-radiation-induced liver disease (RILD [19], defined as a more than two-fold anicteric elevation in alkaline phosphatase with non-malignant ascites), hepatotoxicity (ascites, jaundice, and deterioration of liver function markers such as bilirubin, prothrombin time, aspartate amino transferase, amino alanine transferase, and alkaline phosphatase), and hematotoxicity (any type of anemia). Quality assessment was performed using the Newcastle–Ottawa scale; studies with scores of 7–9 were considered high-quality and those with scores of 4–6 were considered medium-quality [20]. Studies with low quality was discussed to be excluded, referencing the recommendation of the Cochrane Handbook that only observational studies having moderate to low risk of bias possibility should be included in the meta-analyses [21]. Data extraction and quality assessment were performed independently by two researchers; any differences were resolved via mutual discussion.

### 2.3. Statistics

This study was based on the outcome levels of the included studies. Considering the range of clinical variety among patients and institutions and following the recommendation that a random effects model should be default for meta-analyzing studies including non-randomized studies, a random effects model was used for all pooled analyses [22]. Pooled analyses were performed to yield the weighted means of percentiles which corresponded to the primary and secondary endpoints. Since the random effects model is a model that averages the distribution of results affected by chance (i.e., the calculation of statistical heterogeneity is invalid), the heterogeneity between the results was indicated by the pooled estimate and the 95% confidence interval [23]. The visual inspection of funnel plots and quantitative analysis using Egger’s test [24] were used to assess the publication bias, which was examined in pooled analyses that included 10 or more studies. If visual asymmetry was noted and the two-tailed Egger’s test *p*-value was <0.1, Duval and Tweedie’s trim and fill method [25] was performed for sensitivity analysis.

Subgroup analyses were performed using Cochran’s Q-test based on the analysis of variance and mixed effects analyses (i.e., a random effects model to combine studies within each subgroup plus a fixed effect model to combine subgroups and yield the overall effect). Among known clinical indicators, tumor size (with a median tumor size of 5 cm used as a cutoff) and AFP (with a median or mean value of 400 ng/mL or more than half of the patients having AFP levels of >400 ng/mL, used as cutoffs) were used for the subgroup analyses of endpoints; the subgroup analyses of complications were performed considering only tumor size. The cutoff value of tumor size of 5 cm was used, because it has been used in AJCC staging and is associated with tumor aggressiveness such as histologic grade and vascular invasion and patients’ prognosis [26,27,28]. The cutoff value of the AFP level of 400 ng/mL was used, as it was reported that the level was related to oncological outcome and advanced disease in studies in various clinical settings as well [29,30,31]. Radiation dose was not considered a variable in subgroup analyses, because it is commonly influenced by tumor anatomy (e.g., doses are lower for large-size tumors or those near the *porta hepatis*). Meta-regression was performed to evaluate the correlation between local control (in-field failure rate) and intrahepatic- and extrahepatic failure. All statistical analyses were conducted using Comprehensive Meta-Analysis version 3 (Biostat Inc., Englewood, NJ, USA).

### 2.4. Protocol Registration

This study has been registered in PROSPERO (CRD42021273671).

## 3. Results

Our initial search of the databases revealed 311 articles, of which irrelevant article types (reviews, letters, editorials, and conference abstracts) and duplicates were filtered out based on their titles and citations. The abstracts of 163 papers were reviewed, and 143 studies were excluded owing to their irrelevant subjects/formats or insufficient numbers of patients. Full-text reviews were performed for 20 papers, of which 10 were excluded, because they did not fully meet the inclusion criteria. In addition, two other studies were discovered from the reference lists of the selected papers and were included. Hence, 12 studies [32,33,34,35,36,37,38,39,40,41,42,43] involving 757 patients were ultimately included in our analysis; the inclusion process is summarized in Figure 1.

A majority of studies (10 of 12, 83.3%) were retrospectively designed; all were categorized as having medium quality according to the Newcastle–Ottawa scale (Appendix A). Seven of the 12 studies (58.3%) were from South Korea, three were from China and Hong Kong (25%), and one each was from Japan and the United States. The most common etiology of HCC in 10 studies was hepatitis B virus (50–100%), whereas hepatitis C virus was the most common etiology in the studies from Japan (73%) and the United States (51.4%). The rates of female patients ranged from 10% to 33%, with a median value of 21.3%. The PVT rates ranged from 0% to 58.6%, with a median value of 25%. The proportions of patients with Child-Pugh class A ranged from 77.5% to 100%, with a median value of 88.1%. The median tumor sizes ranged from 2.3 to 11.2 cm, with an overall median value of 5.8 cm. In 3 of 10 available studies, more than half of the patients had AFP levels of ≥400 ng/mL. The clinical information of the patients from the included studies is shown in Table 1.

SBRT was the treatment modality in five of 12 studies (41.7%), while a conventional fractionation scheme was used in the remaining seven studies. Using the EQD2 (e.g., equivalent dose calculated in 2 Gy per fraction schedule), the median prescribed doses ranged from 37.3 to 150 Gy, with an overall median of 54 Gy. Three studies recruited patients in the pre-sorafenib era (earlier than 2008), and 17% to 42% of patients received sorafenib in three available studies. No data regarding sorafenib application were provided in the remaining six studies. The clinical information and outcomes of recruited studies are listed in Table 2.

The pooled one- and two-year OS rates were 72.3% (95% confidence interval: 60.2–81.9%) and 50.5% (95% CI: 35.6–65.4%), respectively. The pooled complete and overall response rates were 15.9% (95% CI: 9.2–36.3%) and 72.2% (95% CI: 65.4–78.1%), respectively. The pooled in-field, intrahepatic, and extrahepatic recurrence rates were 13.4% (95% CI: 8.8–19.9%), 45.6% (95% CI: 37.9–53.4%), and 26.6% (95% CI: 20.3–34.0%), respectively. On subgroup analyses, the one-year OS rates were significantly different according to the median tumor size (≥5 vs. <5 cm: 62.4% vs. 82.8%; *p* = 0.036) and the AFP level (the median or 50% of patients ≥400 vs. <400 ng/mL: 40.5% vs. 71.7%; *p* < 0.001). The two-year OS rates were also significantly different according to the median tumor size (≥5 vs. <5 cm: 41.8% vs. 69.6%; *p* = 0.011) and the AFP level (≥400 vs. <400 ng/mL: 22.7% vs. 51.9%; *p* < 0.001). The complete response rates were also significantly different according to the median tumor size (≥5 vs. <5 cm: 11.7% vs. 37.7%; *p* < 0.001) and the AFP level (≥400 vs. <400 ng/mL: 5.7% vs. 21.9%; *p* = 0.004). The extrahepatic recurrence rates were significantly different according to the tumor size (≥5 vs. <5 cm: 33.5% vs. 19.5%; *p* = 0.01). On toxicity analyses, the pooled rates of grade ≥3 gastrointestinal toxicity, RILD, hepatotoxicity, and hematotoxicity were 5.0% (95% CI: 3.0–8.2%), 3.5% (95% CI: 1.4–8.4%), 5.7% (95% CI: 3.1–10.5%), and 4.9% (95% CI: 2.3–10.0%), respectively. Subgroup analyses according to the tumor size revealed no significantly difference.

Possible publication biases were noted in the pooled analyses of one-year OS, complete response, overall response, in-field failure, gastrointestinal complication, and hepatotoxicity rates. The detailed results of all pooled analyses, heterogeneity analyses, publication bias and sensitivity analyses (using Duval and Tweedie’s methods) are shown in Table 3 and Table 4.

Finally, meta-regression analyses were performed to evaluate the relationships between local control (in-field failure rate) and intrahepatic and extrahepatic metastases. We found no significant relationships, as the *p*-values were very high (*p* = 0.9058 between the in-field failure and the intrahepatic failure rate (Figure 2A) and *p* = 0.8748 between the in-field failure and the extrahepatic failure rate (Figure 2B)).

## 4. Discussion

Incomplete TACE commonly occurs owing to anatomical hindrances. TACE is less efficacious against large tumors, because the inner core of the mass becomes avascularized and hypoxic; hence, arterial-directed therapy has limited efficacy (Figure 3A) [44]. The presence of collateral feeding vessels for tumors in certain locations (e.g., segment 4 or the caudate lobe) [45,46] or with portal venous obstruction is a hindrance to complete TACE (Figure 3B) [47]. Given that EBRT can be applied regardless of the tumor location and it can achieve the desired dose distribution across the entire tumor, it is highly advantageous in terms of overcoming these anatomical challenges (Figure 3A,B) [11]. It is not surprising that, in a recent South Korean survey involving 162 physicians, EBRT (67.7%) and sorafenib (70.2%) were the two most common treatments of choice after incomplete TACE despite the lower evidence level for EBRT in the literature [48,49].

In the present study, the clinical indicators of tumor size and biological aggressiveness, which were reflected by the AFP level, were significant factors affecting survival and complete response rates. The majority of patients recruited from the included studies had locally advanced diseases with a high prevalence of such indicators. The overall median tumor size was 5.8 cm; the median in three studies exceeded 10 cm. The median PVT rate was 25%, and approximately 40% to 50% of all the recruited patients had AFP levels higher than 400 ng/mL. Despite these characteristics, EBRT showed favorable efficacy and feasibility, as reflected by a high local control rate (86.6%) (pooled in-field failure rate: 13.4%); the pooled grade 3 toxicity rates ranged from 3.5% to 5.7%.

In comparison, some authors have reported data from patients refractory to TACE who underwent sorafenib administration or who repeated TACE (deemed conventional approaches). In a study by Ogasawara et al. [50], sorafenib administration yielded superior oncologic outcomes than repeating TACE with respect to OS (median: 25.4 vs. 11.5 months; *p* = 0.003) and liver dysfunction (median time to liver dysfunction: 29.8 vs. 17.0 months; *p* = 0.003). Three patients (15%) discontinued sorafenib owing to severe toxicities. Of note is that, since this study mainly targeted HCC patients with intermediate stage of Barcelona Clinic of Liver Cancer (BCLC) system [51], the OS was favorable than that of sorafenib studies that included patients with advanced stage (BCLC C or higher) [9,10] as the main target. A recent Chinese study produced similar results in that sorafenib administration resulted in a better OS rate than repeating TACE (median OS: 17.9 vs. 7.1 months; *p* < 0.001) [52]; two patients (6.7%) experienced grade 3 hand–foot syndrome. The partial response and stable disease rates in the sorafenib arms were 5% and 70%, respectively, in the former study and 16.7% and 56.7%, respectively, in the latter. Considering the pooled two-year OS rate (50.5%) and response rate (72.2%) in our study, salvage EBRT produces survival outcomes that are at least comparable to sorafenib administration with higher tumor response rates. The recent randomized study by Yoon et al. [53], of which the reported benefits regarding the OS and the progression-free survival of TACE plus EBRT as compared to those of sorafenib, supports the use of the combined treatment as well. Since the overwhelming majority of patients in our meta-analysis did not receive sorafenib (e.g., patients recruitment in the pre-sorafenib era) or other systemic agents, the combined or sequential use of systemic agents with EBRT should be investigated for the possibility of further improving clinical outcomes.

In the present study, the pooled intrahepatic recurrence rate was as high as 45.6%, while the extrahepatic recurrence rate was 26.6%. Of note, local control (intra-hepatic failure) did not correlate with intrahepatic or extrahepatic recurrences (the *p*-values were very high at 0.6341 and 0.8529, respectively), suggesting that local control might not be sufficient to significantly impede non-local progression. The high incidences of non-local recurrences, despite efficient local control, necessitate systemic treatment. Although sorafenib has been a standard systemic treatment for HCC, its oncologic efficacy is not fully satisfactory, with a response rate of <3% and a gain of time to progression of <3 months [9,10]. Immunotherapy can be introduced to reduce these non-local recurrences, because interactions between tumor and immune cells during chronic liver inflammation create an environment that favors tumor progression [54]. In a recent phase 3 trial, the combined administration of atezolizumab (a programmed death-1 ligand inhibitor) and bevacizumab (a vascular endothelial growth factor inhibitor) had superior outcomes than sorafenib administration [55]; the one-year OS benefit was significant (67.2% vs. 54.6%; *p* < 0.001), progression-free survival was 2.5 months longer (*p* < 0.001), and the objective response rate was better (27.3% vs. 11.9%). Until recently, 2nd-line agents such as regorafenib, ramucirumab, and carbozantinib have been developed, and lenvatinib as a first-line agent shows a non-inferior effect to sorafenib, but these are the first to show superior results than sorafenib as a first-line agent [56,57]. Recent National Cancer Comprehensive Network guidelines recommend the atezolimumab and bevacizmab combination as the preferred systemic agent for advanced HCC [58]. Therefore, we strongly recommend clinical trials investigating immunotherapy plus EBRT for patients with incomplete TACE to assess the possibility of reducing non-local failure and providing other oncologic benefits.

The main limitation of our meta-analysis was that out study included mostly non-randomized observational studies and heterogenity among studies, which might affect pooled aresults. However, incomplete response after TACE is a not uncommon clinical situation, which neccesitate further clinical decisions. Many clinical decisions inevitably depend on information from observational studies, especially in treating intractable situations such as those after incomplete TACE [59,60]. The meta-analysis of observational studies might be one of a few available methods to integrate the literature and yield information for clinical decision making.

## 5. Conclusions

Our study demonstrated the efficacy and feasibility of EBRT after incomplete TACE. Although a significant proportion of the patients had locally advanced tumors and aggressive biology (as reflected by AFP levels), the pooled in-field failure rate was only 13.4% and the rate of serious complications was acceptable. The pooled rates of RILD and hepatotoxicity, which are of concern in patients undergoing EBRT for locally advanced HCCs, were only 3.5% and 5.7%, respectively. However, efficient local control did not improve intrahepatic out-of-field control or reduce extrahepatic recurrences. The administration of the selected immunotherapeutic agents showed promising results in recent studies, with meaningful response rates and longer intervals before progression. In conclusion, our data suggest that EBRT can serve as a standard salvage option for incomplete TACE and warrant future studies investigating the combined approach of modern immunotherapy and EBRT.

## Figures and Tables

**Figure 1 medicina-57-01000-f001:**
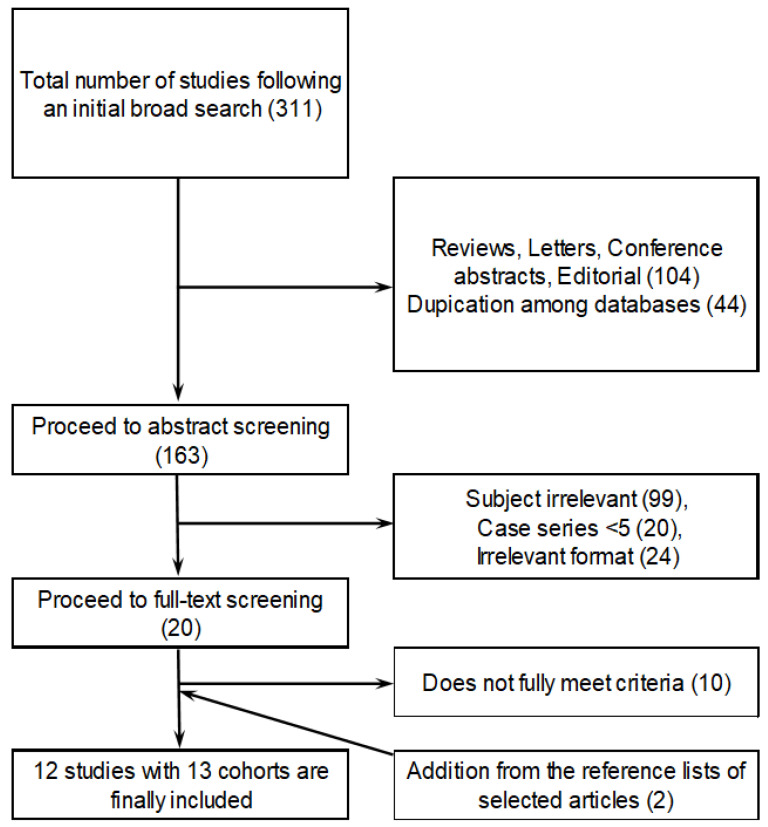
Flowchart showing the study selection process.

**Figure 2 medicina-57-01000-f002:**
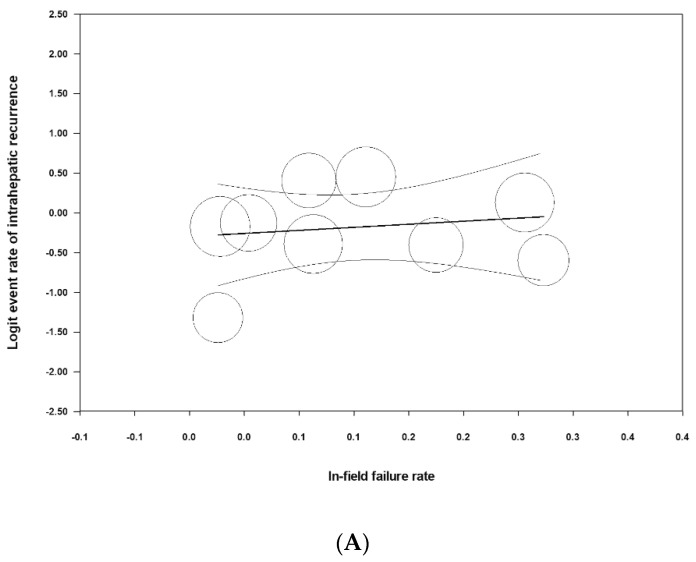
Scatterplots of meta-regression analyses: (**A**) comparison between the logit event rates of intrahepatic recurrence and in-field failure (*p* = 0.6341); (**B**) comparison between the logit event rates of intrahepatic recurrence and extrahepatic failure (*p* = 0.8529).

**Figure 3 medicina-57-01000-f003:**
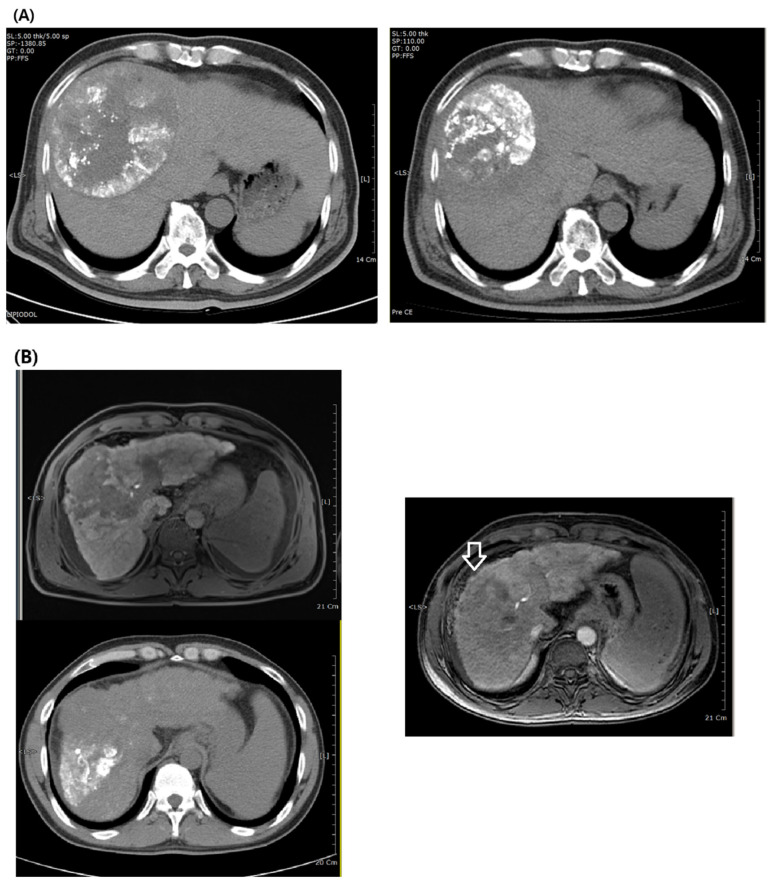
(**A**) **Left panel:** A 59-year old man diagnosed with hepatitis B virus (HBV)-related hepatocellular carcinoma on segment 8; the diameter of the largest lesion was 13.4 cm. Lipiodol tagging revealed that two rounds of transarterial chemoembolization (TACE) were insufficient, especially at the tumor’s core; **right panel:** four months after external radiotherapy (40 Gy/16F) and a third round of TACE. The tumor size was significantly decreased, and the efficiency of TACE was enhanced as evidenced by a higher level of lipiodol tagging inside the tumor. (**B**) **Left upper panel:** A 42-year old man diagnosed with HBV-related hepatocellular carcinoma within a cirrhotic liver. The tumor extended around segment 4 and was accompanied by a left main portal vein thrombus. **Left lower panel:** The initial TACE only produced physiologic distribution of lipiodol and poor tumor tagging; **right panel:** one month after EBRT (54 Gy/27 F), the tumor and accompanying right portal vein thrombus significantly decreased (white arrowhead: normalized outline of the liver which the tumor previously extruded).

**Table 1 medicina-57-01000-t001:** Selected published studies with correlating subjects.

First Author	Affiliation	Country	Inclusion Period	Study Design	*n*	Etiology	Age	Female (%)	Performance Status	AFP of ≥400 ng/mL (%)	PVT	CPC A	Tumor Size
Oh [33]	Samsung hospital	Korea	2006–2007	P	40		M59.5 (36–92)	22.5%	0–1 (97.5%)	38.5%	25.0%	90.0%	≥5 cm (55%)
Kim [32]	National Cancer Center	Korea	2001–2005	R	70	HBV 82.9%	M57 (30–78)	10.0%	0–2 (100%)	62.7%	58.6%	88.1%	M7.5 cm (2–17)
Choi [28]	Multicenter (prospective)	Korea	2008–2011	P	31	HBV 77.4%	M63.2 (36–74)	19.4%	0 (77.4%), 1 (22.6%)	38.7%	* 29%	96.8%	M6.6 cm (5.1–17)
Kang [30]	Korea Institute of Radiological and Medical Science	Korea	2008–2011	R	47	HBV 68%		21.3%	0–1 (100%)		10.6%	87.2%	M2.9 cm (1.3–8)
Shim [35]	Yonsei Cancer Center	Korea	1992–2002	R	38		M53 (38–79)	15.8%	0–1 (86.8%)	47.4%	* 31.6%	86.8%	M10.2 (5–17)
Zhong [37]	Fuzhou General Hospital	China	2006–2012	R	72	HBV 75.7%	~M52.5	20.8%	0–1 (75%), 2 (25%)	81.9%	NA	NA	≥10 cm only
Chiang [27]	Tuen Mun Hospital	Hong Kong	2008–2015	R	72	HBV 84.7%	M60 (28–87)	15.0%	0 (71%),	M893.5 (ng/mL)	* 25%	100%	M 11.2 cm (5–23.6)
									1 (6%),				
									2 (22%)				
Jacob [29]	Univ. of Birmingham	United	2008–2013	R	37	HCV 51.4%; Alcohol 18.9%	Mean 64.4	27.2%		Mean 32.7	NA	Mean score 6.3 ± 1.2	Mean 6.1 cm ± 2.4
		States											
‡ Kibe [31]	Ofuna Chuo Hospital	Japan	2005–2017	R	144	HCV 73%	M73 (40–89)	33.0%			BCLC C 28%	90.3%	M2.3 cm (1–6.2)
† Yao [36]	Guangxi Traditional Chinese Medicine University	China	2008–2015	R	33	HBV 100%	M55 (42–75)	24.2%	All KPS ≥70	15.2%	0%	100%	mean PTV 128 cm^3^
Byun (high dose) [26]	Yonsei Cancer Center	Korea	2001–2016	R	62	HBV 69.4%; HCV 16.1%	M68 (37–83)	24.2%	0–2 (100%)	M21.1 (ng/mL)	21.0%	87.1%	M3 cm (1–20)
Byun (low dose) [26]					62	HBV 61.3%; HCV 17.7%	M68 (41–84)	25.8%		M18.0 (ng/mL)	19.4%	82.3%	M4 cm (1–15)
Park [34]	Korea University Ansan Hospital	Korea	2010–2019	R	40	HBV 62.5%	M60 (43–77)	17.5%	0 (33%); 1 (65%)	22.5%	30% (main PVT 25%)	77.5%	M3.4 cm (0.8–20)
						NBNC 22.5%							

Abbreviations: AFP, alpha-fetoprotein; PVT, portal vein thrombosis; CPC, Child-Pugh class; RTx., radiotherapy; OS, overall survival; CR, complete remission; PR, partial response; 3DCRT, 3-dimensional conformal radiotherapy; SBRT, stereotactic body radiotherapy; HFRT, hypofractionated radiotherapy; HBV, hepatitis B virus; HCV, hepatitis C virus; NBNC, negative for hepatitis B surface antigen and hepatitis C antibody; KPS, Karnofsky performance status. Heading capital M denotes the median value; * excludes main PVT; ‡ SBRT for residuals after radiofrequency ablation or surgery as well as transarterial chemoembolization (TACE); † Recurrent as well as residual after TACE.

**Table 2 medicina-57-01000-t002:** Clinical outcomes of the included studies.

First Author	RTx (Dose per Fraction)	Median † EQD2_10 Gy_	Sorafenib during Follow-Up	Follow-Up (Months)	OS	CR/PR	Pattern of Failure			m/c EHM Site	Grade ≥3 Toxicity				Grade 5 Toxicity
							In-Field	Out-Field, Intrahepatic	Extrahepatic		GI	RILD	Hepatic	Hemato-Logic	
Oh	M54 Gy, 3DCRT (3 Gy)	58.5	Pre-sorafenib era	M17.8	M19 months, 72.0% and 45.6% for 1- and 2-year OS rates, respectively	20.9%; 41.9%	22.5%	40.0%	32.5%		0.0%	12.5%	0.0%	0.0%	None
Kim	M54 Gy, 3DCRT (2–3 Gy)	54	Pre-sorafenib era	M8.8	M10.8 months; 43.1% and 17.6% for 1- and 2-year OS rates, respectively	5.7%; 48.6%	2.8%	45.7%	35.7%	Lung	12.9%	NA	5.7%	NA	None
Choi	M54 Gy, 3DCRT (1.8–2 Gy)	54	NA	M30	61.3% and 61.3% for 1- and 2-year OS rates, respectively	12.9%; 64.5%	32.3%	35.5%	41.9%	Lung	0.0%	0.0%	22.6%	12.9%	None
Kang	Up to 60 Gy/3 F, SBRT	Up to 150	NA	M17	86.4% and 68.7% for 1- and 2-year OS rates, respectively	38.3%; 38.3%	5.4% (2-year)	46.8%	21.3%	Lung	10.6%	NA	8.6%	10.6%	None
Shim	Mean 54 Gy, 3DCRT (1.8 Gy)	53.1	Pre-sorafenib era		65.8% and 36.8% for 1- and 2-year OS rates, respectively	0%; 65.8%	2.6%	21.1%			0%	NA	13.2%	0.0%	None
Zhong	35.6 Gy; (2.6–3 Gy) HFRT	37.8	NA	M18	M12.2 months; 38% and 27.8% for 1- and 2-year OS rates, respectively	8.3%; 70.8%					0.0%	0.0%	0.0%	0.0%	None
Chiang	30–39 Gy/6 F or 24–40 Gy/6–10 F SBRT	37.3	17.5%	M16.8	M19.9 months	0%; 68%	16.1% (2-year)	61.1%	27.7%		2.8%	0.0%	4.2%	NA	1 case
Jacob	45 Gy/3 F, SBRT	93.8	41.9%		M33 months; 81.1% and 67.6% for 1- and 2-year OS rates, respectively	30.3%; 57.6%	10.8%				2.7%	0.0%	0.0%	0.0%	None
Kibe	35 or 40 Gy/5F, SBRT	60	NA	M34.8	95.1, 79.6, 66.1% (1-, 2-, and 3-year)		11.1%				0.0%	0.0%	0.0%	NA	None
Yao	39–45 Gy/3-5F, SBRT		NA		M19 months, 75.8% and 45.5% (1- and 2-year OS rates, respectively	18.9%; 56.9%					3.0%	NA	6.1%	3.0%	None
Byun (high dose)	M60 Gy (2–6 Gy) ‡ conventional, SBRT	65.1	NA	M14.2	75.8% (1-year)		11.3%	40.3%	17.7%	Lung	3.2%	5.3%	NA	NA	None
Byun (low dose)	M50 Gy (1.8–5 Gy) ‡ conventional, SBRT	49.6			62.9% (1-year)		30.6%	53.2%	11.3%	Lung	* (*n* = 261) 6.1%	13.8%	NA	NA	None
Park	M40 ‡ conventional, SBRT	47.8 (conventional) 57 (SBRT)	32.5%	M14.4	82.2% and 55.8% for 1- and 2-year OS rates, respectively	37%; 41.3%	10.9% (2-year)	60.0%	30.0%	Lung	0.0%	0.0%	5.0%	5.0%	None

Abbreviations: RTx., radiotherapy; OS, overall survival; CR, complete remission; PR, partial remission; EHM, extrahepatic metastasis; GI, gastrointestinal; RILD, radiation-induced liver disease; 3DCRT, 3-dimensional conformal radiotherapy; SBRT, stereotactic body radiotherapy; HFRT, hypofractionated radiotherapy † Equivalent dose in 2 Gy per fraction scheme with an α/β ratio of 10; uppercase M denotes the median value. ‡ 3DCRT of the intensity-modulated radiotherapy performed in 1.8 or 2 Gy per fractions. * Complication data was provided from original unmatched population of 261 patients (e.g., other clinical outcomes were from matched 62 patients).

**Table 3 medicina-57-01000-t003:** Pooled analysis of oncologic outcomes.

	Subgroups	Cohorts (*n*)	Patients (*n*)	Events % (95% CI)	*p*, Subgroup Difference	Egger’s Test, *p*	Trimmed Value ^a^
*Overall survival*							
1-year OS							
	All studies	12	485	72.3 (60.2–81.9)		0.002, no change	
	Tumor size of ≥5 cm	7	321	62.4 (48.6–74.5)	0.036		
	Tumor size of <5 cm	5	355	82.8 (68.0–91.7)			
	^b^ High AFP level (≥400 ng/mL)	2	142	40.5 (32.8–48.8)	<0.001		
	Low AFP level (<400 ng/mL)	8	343	71.7 (65.8–76.9)			
2-year OS							
	All studies	10	552	50.5 (35.6–65.4)		0.252	
	Tumor size of ≥5 cm	7	321	41.8 (28.7–56.2)	0.011		
	Tumor size of <5 cm	3	231	69.6 (53.7–81.8)			
	^b^ High AFP level (≥400 ng/mL)	6	219	22.7 (14.2–34.3)	<0.001		
	Low AFP level (<400 ng/mL)	2	142	51.9 (42.7–61.1)			
*Response rate*							
Complete response							
	All studies	10	480	15.9 (9.2–36.3)		0.004, 19.4 (11.2–31.6)	
	Tumor size of ≥5 cm	8	393	11.7 (6.3–20.7)	<0.001		
	Tumor size of <5 cm	2	87	37.7 (28.2–48.3)			
	^b^ High AFP level (≥400 ng/mL)	3	214	5.7 (2.5–12.7)	0.004		
	Low AFP level (<400 ng/mL)	6	219	21.9 (13.7–33.2)			
Overall response							
	All studies	10	480	72.2 (65.4–78.1)		0.018, 68.1 (60.8–74.5)	
	Tumor size of ≥5 cm	8	393	71.0 (62.8–78.0)	0.295		
	Tumor size of <5 cm	2	87	77.4 (67.4–85.0)			
	^b^ High AFP level (≥400 ng/mL)	3	214	67.6 (52.3–79.9)	0.41		
	Low AFP level (<400 ng/mL)	6	219	74.2 (66.0–81.0)			
*Failure pattern*							
In-field failure							
	All studies	11	643	13.4 (8.8–19.9)		0.043, 17.2 (11.3–25.3)	
	Tumor size of ≥5 cm	6	288	13.2 (6.7–24.1)	0.989		
	Tumor size of <5 cm	5	355	13.1 (7.0–23.1)			
	^b^ High AFP level (≥400 ng/mL)	2	142	7.7 (1.3–34.5)	0.359		
	Low AFP level (<400 ng/mL)	7	310	17.0 (10.4–26.7)			
Intrahepatic recurrence							
	All studies	9	462	45.6 (37.9–53.4)			
	Tumor size of ≥5 cm	5	251	41.2 (28.9–54.8)	0.304		
	Tumor size of <5 cm	4	211	49.5 (41.4–57.6)			
	^b^ High AFP level (≥400 ng/mL)	2	142	53.5 (38.4–67.9)	0.228		
	Low AFP level (<400 ng/mL)	6	273	42.0 (31.9–52.8)			
Extrahepatic recurrence							
	All studies	8	424	26.6 (20.3–34.0)			
	Tumor size of ≥5 cm	4	213	33.5 (27.4–40.1)	0.01		
	Tumor size of <5 cm	4	211	19.5 (13.1–28.1)			
	^b^ High AFP level (≥400 ng/mL)	2	142	31.8 (24.5–40.1)	0.35		
	Low AFP level (<400 ng/mL)	5	235	25.2 (15.9–37.4)			

Abbreviations: AFP, alpha-fetoprotein; CI, confidence interval; OS, overall survival. ^a^ Modified value after using Duval and Tweedie’s trim and fill to evaluate possible publication bias. ^b^ The median or mean value is higher than 400 ng/mL, or more than half of the patients were in a subgroup with the AFP levels of >400 ng/mL.

**Table 4 medicina-57-01000-t004:** Pooled analysis of grade ≥3 complications.

Subgroups	Cohort (*n*)	Patients (*n*)	Events % (95% CI)	*p*, Subgroup Difference	Egger’s Test, *p*	Trimmed Value ^a^
Gastrointestinal						
All studies	13	947	4.1 (2.4–7.0)		0.001, 6.0 (3.6–10.1)	
Tumor size of ≥5 cm	8	393	3.1 (1.2–7.6)	0.503		
Tumor size of <5 cm	5	554	4.7 (2.2–9.9)			
RILD						
All studies	9	759	3.5 (1.4–8.4)			
Tumor size of ≥5 cm	5	252	2.3 (0.5–10.3)	0.571		
Tumor size of <5 cm	4	507	4.1 (1.0–14.9)			
Hepatotoxicity						
All studies	11	624	5.7 (3.1–10.5)		0.001, 8.5 (4.5–15.5)	
Tumor size of ≥5 cm	8	393	6.3 (3.0–12.8)	0.572		
Tumor size of <5 cm	3	231	4.0 (1.0–14.7)			
Hematotoxicity						
All studies	8	338	4.9 (2.3–10.0)			
Tumor size of ≥5 cm	6	251	2.9 (0.9–8.9)	0.116		
Tumor size of <5 cm	2	87	8.5 (4.1–16.8)			

Abbreviations: CI, confidence interval; RILD, radiation-induced liver disease. ^a^ Modified value after using Duval and Tweedie’s trim and fill method to evaluate possible publication bias.

## Data Availability

All data and materials are provided in the manuscript and included data.

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
