# Peer review of "Salvage External Beam Radiotherapy after Incomplete Transarterial Chemoembolization for Hepatocellular Carcinoma: A Meta-Analysis and Systematic Review"

_medicina, 2021, doi:10.3390/medicina57101000_

Round 1

Reviewer 1 Report

To the authors:

“Salvage external radiotherapy after incomplete transarterial chemoembolization for hepatocellular carcinoma: A meta-analysis and systematic review”

My comments are as follows.

  1. Please show the rationale for setting the cutoff value of AFP to 400 ng/mL.
  2. Please show the rationale for setting the cutoff value of tumor size to 5 cm.
  3. I recommend changing “external radiotherapy” to “external beam radiotherapy” in Title.
  4. You mentioned “EBRT was feasible and effcient in regards of tumor response and control; after incomplete TACE”. As you know, drug therapy and hepatic arterial infusion chemotherapy (HAIC) are treatment options for advanced HCC. How do you think about the proper use of drug therapy, HAIC or EBRT after incomplete TACE? Please explain your opinion.

I hope that my comments will be useful in improving the article.

Reviewer 2 Report

You should be discussed in the discussion with the addition of a paper showing that EBRT and TACE were more effective than Sorafenib (yoon 2018)

How does it compare to EBRT for atezolizumab + bevacizumab?
After sorafenib, regorafenib, ramucirumab and cabozantinib treatment is recommended, how does it compare with EBRT?

The section discussing the efficacy of sorafenib in disccusion is inconsistent (lines 260-262 and 281-282)

Round 2

Reviewer 1 Report

Comments to Author: The manuscript has been revised well.

Author Response

We appreciate your kind comment. 

Reviewer 2 Report

Thank you for your thoughtful and clear reply.
Since there are few parts added to the text, I think it would be better to add to the text the parts you answered in detail in your comment. This would make it easier for readers to understand.

Author Response

Thank you for your considerate comment.

Agreeing the reviewer, we added some discussion about the reason of outcome differences between Ogasawara et al. study and pivotal studies of sorafenib (line 280-284) and the brief review of chemotherapy for HCC and recent NCCN guidelines about atezolimumab and bevacizumab combination (line 313-318).